# Transposon-modified antigen-specific T lymphocytes for sustained therapeutic protein delivery in vivo

Richard T. O'Neil[1,2], Sunandan Saha[3], Ruth Ann Veach[1,2], Richard C. Welch[1,2], Lauren E. Woodard[1,2,4], Cliona M. Rooney[5,6,7,8] & Matthew H. Wilson[1,2,4]

A cell therapy platform permitting long-term delivery of peptide hormones in vivo would be a significant advance for patients with hormonal deficiencies. Here we report the utility of antigen-specific T lymphocytes as a regulatable peptide delivery platform for in vivo therapy. *piggyBac* transposon modification of murine cells with luciferase allows us to visualize T cells after adoptive transfer. Vaccination stimulates long-term T-cell engraftment, persistence, and transgene expression enabling detection of modified cells up to 300 days after adoptive transfer. We demonstrate adoptive transfer of antigen-specific T cells expressing erythropoietin (EPO) elevating the hematocrit in mice for more than 20 weeks. We extend our observations to human T cells demonstrating inducible EPO production from Epstein–Barr virus (EBV) antigen-specific T lymphocytes. Our results reveal antigen-specific T lymphocytes to be an effective delivery platform for therapeutic molecules such as EPO in vivo, with important implications for other diseases that require peptide therapy.

[1] Division of Nephrology and Hypertension, Department of Medicine, Vanderbilt University School of Medicine, Nashville, TN 37232, USA. [2] The Vanderbilt Center for Kidney Disease, Vanderbilt University School of Medicine, Nashville, TN 37232, USA. [3] Interdepartmental Program in Translational Biology and Molecular Medicine, Baylor College of Medicine, Houston, TX 77030, USA. [4] Department of Veterans Affairs, Nashville, TN 37212, USA. [5] Center for Cell and Gene Therapy, Baylor College of Medicine, Houston, TX 77030, USA. [6] Department of Pediatrics, Baylor College of Medicine, Houston, TX 77030, USA. [7] Department of Immunology, Baylor College of Medicine, Houston, TX 77030, USA. [8] Department of Molecular Virology and Microbiology, Baylor College of Medicine, Houston, TX 77030, USA. Correspondence and requests for materials should be addressed to M.H.W. (email: matthew.wilson@vanderbilt.edu)

The development of a cell therapy platform for safe and long-term delivery of peptide hormones in vivo would be a significant advance for patients with a variety of hormonal deficiencies. T lymphocytes are promising candidates for peptide hormone delivery platforms because they can be harvested relatively easily by phlebotomy, efficiently genetically modified ex vivo, stored for future use, and they can enter the memory compartment and can be sustained for many years[1]. Adoptively transferred T lymphocytes have recently been embraced as a promising therapeutic platform in oncology. A prerequisite for cell-based adoptive transfer therapy is survival and engraftment of the therapeutic cells, processes that are augmented in the presence of cognate antigen[2]. T lymphocytes specific for antigens presented by latent viral infections such as Epstein–Barr virus (EBV) persist for many years after adoptive transfer[3, 4]. Vaccination can be used to boost genetically modified lymphocytes expressing protein hormones[5]. For these reasons, antigen-specific T cells, such as EBV-specific T lymphocytes, may represent a useful platform for sustained systemic hormone delivery.

Currently, therapeutic protein delivery requires providing recombinant protein, which often differs in structure from the protein made in vivo and is costly to administer often requiring repeated injections or infusions[6]. One example of this is erythropoietin (EPO), which is a peptide hormone that regulates red blood cell production[7]. Gene and cell therapy for sustained production of EPO in situ represents a model system for evaluating therapeutic protein production in vivo as one can evaluate hematocrit as a readout of EPO production. Researchers have reported viral vector-based strategies for transduction of muscular, hepatic, or dermal tissue with constructs driving EPO production[8–12]. Although these strategies increased hemoglobin concentration, viral vector-based approaches have inherent drawbacks related to their immunogenicity, limited control of EPO production afforded by viral construct packaging restraints, and difficulty in reversing the procedure, which may require surgical removal of transduced tissue in cases of EPO over production. In the current studies, we evaluated a non-viral transposon-based approach for ex vivo engineering T lymphocytes to produce EPO while aiming to circumvent some of the limitations associated with viral vector-mediated gene-based approaches.

Previous studies have established the utility of non-viral transposon systems such as piggyBac for efficient T-cell genome modification[13]. Several features of transposon systems make them attractive tools for generating cell therapy platforms, including potentially reduced immunogenicity compared to viral vectors and capacity for multi-gene insertion that is facilitated by the relatively large cargo capacity and ability to deliver multiple constructs to a single cell[14]. Another transposon system, Sleeping Beauty, is currently approved for use in human clinical trials aimed at engineering T cells to target CD19-positive B-cell malignancies for immunotherapy[15].

We sought to leverage a non-viral transposon system to develop an efficient T-cell-based platform for in vivo delivery of therapeutic peptides such as EPO. In these studies, we demonstrate the feasibility of using antigen-specific CD8+ T lymphocytes to deliver EPO long term in an animal model, correct anemia in an anemic animal model of kidney disease, and provide data supporting the use of EBV-specific T lymphocytes for regulated EPO expression from human T cells.

## Results

**Transposon modification of mouse T lymphocytes**. We engineered a series of piggyBac vectors for genetic modification of T cells to enable tracking of lymphocytes, quantitation of their persistence in vivo, and to express both murine and human EPO (Fig. 1). We first genome-modified murine CD8+ lymphocytes with the pT-effluc-thy1.1 transposon, confirmed luciferase expression from transferred cells by bioluminescent imaging, and observed thy1.1 expression by flow cytometry. We routinely observed that ~35% of the cells were transgene positive after 24 h of in culture (Fig. 2a).

**Plasmid vaccine boosts adoptively transferred T cells**. Luciferase modification enabled us to track T cells in vivo, quantitate their ability to persist long term after adoptive transfer, and to evaluate various vaccination strategies to identify those that would best produce long-term cell engraftment and transgene expression. To this end, we used OT-1 T cells that express a transgenic T-cell receptor (TCR) for a peptide fragment derived from chicken ovalbumin (SIINFEKL) presented on H2-K$^b$ major histocompatibility complex (MHC) class I[16], and SIINFEKL as the vaccine antigen. A plasmid-based vaccination strategy designed to provide efficient antigen presentation to transferred lymphocytes was evaluated.

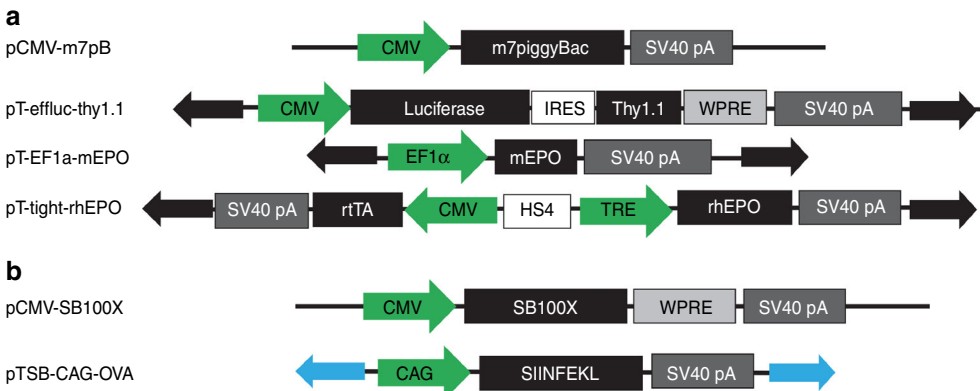

**Fig. 1** Vector schematics. **a** The piggyBac transposase was used with the pT-Tight-hEPO, pT-EF1α-mEPO, and pT-effluc-Thy1.1 transposons. **b** The Sleeping Beauty transposase was used with the pTSB-CAG-OVA transposon. CMV, cytomegalovirus immediate early enhancer/promoter; piggyBac, transposase; pA, SV40 polyadenylation signal; HS4E, core insulator sequence from the chicken B-globin 5′HS4 element; mEPO, recombinant murine erythropoietin cDNA; hEPO, recombinant human erythropoietin cDNA; CAG, CAG synthetic promoter sequence; effluc, enhanced firefly luciferase; Thy1.1, mouse thy1.1 antigen; TRE, tetracycline response element; Tet-ON 3G, tetracycline transactivator; WPRE, woodchuck hepatitis post-transcriptional regulatory element; IRES, internal ribosomal entry site; SV40, simian virus 40 late polyadenylation signal. Solid arrows indicate inverted terminal repeat (ITR) sequences: black, piggyBac ITRs; blue, Sleeping Beauty ITRs

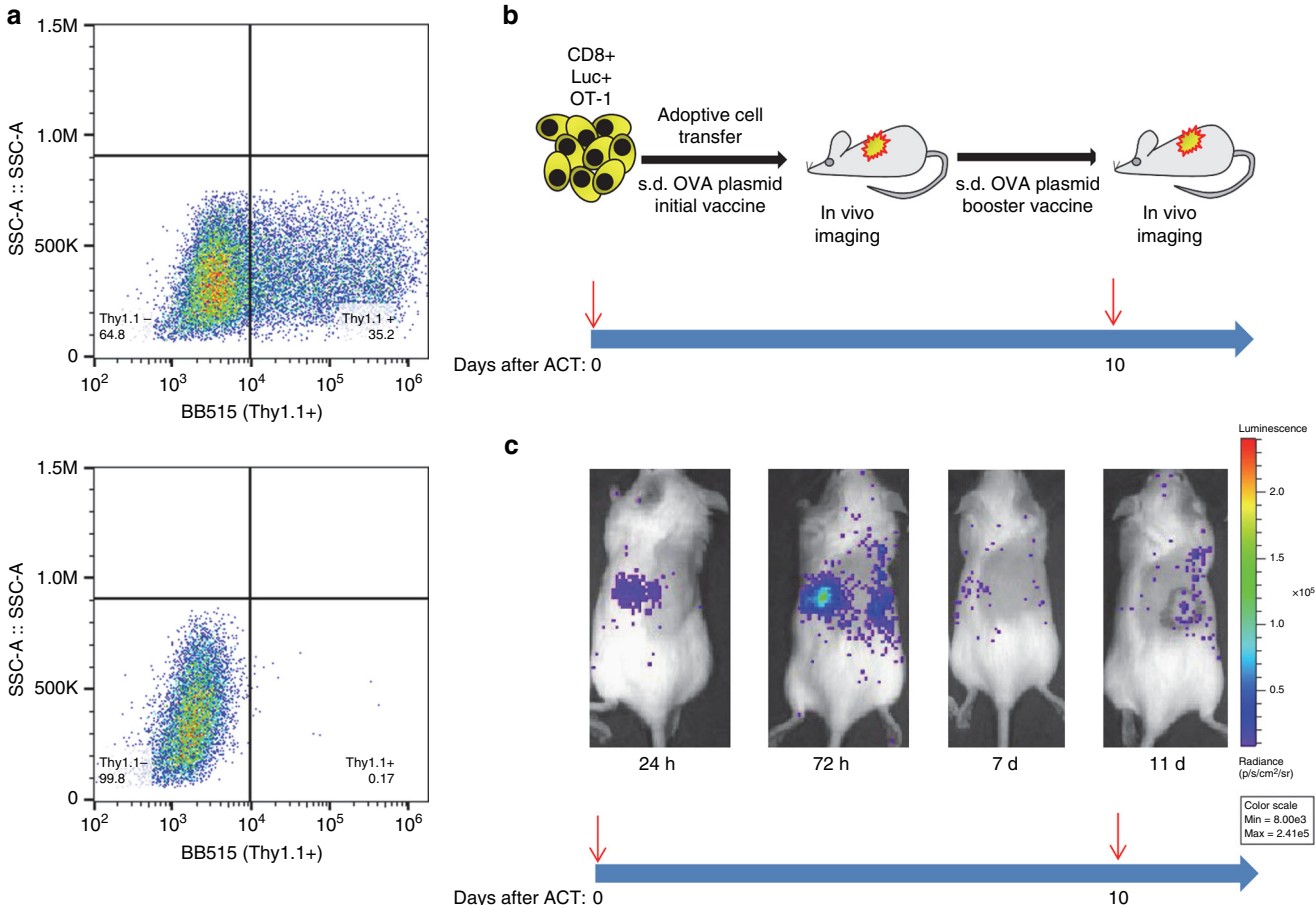

**Fig. 2** Transposon modification and functional engraftment of OT-1 T cells. CD8+ T cells were modified with the pT-effluc-thy1.1 transposon, and 1×10⁷ CD8+ T cells were transferred into host mice. **a** Representative flow cytometry analysis (from $N = 3$) showing ~35% of T cells transfected with pT-effluc-thy1.1 plus pCMV-m7pB were thy1.1-positive after 24 h culture post transfection (top) compared to cells transfected with pTSB-CAG-OVA plus pCAG-SB100X (negative control; bottom). **b** Schematic representation of experimental design. **c** Representative bioluminescent imaging ($N = 5$) from a mouse subdermally injected with the pTSB-CAG-OVA plasmid vaccine immediately after adoptive cell transfer (ACT) and on day 10 after ACT. The blue arrow timeline indicates the days after ACT. Red arrows indicate vaccine administration immediately after ACT and on day 10 after ACT. The images **a**–**c** correspond to the blue arrow timeline

To assess the response to vaccination we adoptively transferred pT-effluc-thy1.1 transposon-modified CD8+ cells, and tracked engraftment by in vivo bioluminescence imaging. Transposon vaccines have been shown to produce sustained antigen expression and improved antigen-specific T-cell responses in vivo[17]. We chose the *Sleeping Beauty* system for our vaccine, to avoid inducing an immune response to the *piggyBac* transposase, which was used for T-cell modification to enable long-term transgene expression. We initially tested subdermal (s. d.) route for vaccine delivery by injecting a plasmid mixture containing pTSB-CAG-OVA transposon and the hyperactive pCMV-SB100X transposase (Fig. 1), complexed with in vivo-jetPEI transfection reagent into the flank of a C57/Bl6 mice immediately after infusion of OT-1 CD8+ T cells (Fig. 2b). We observed recruitment of adoptively transferred luciferase positive cells to the vaccine site (Fig. 2c). This response was transient with a peak response at 72 h followed by luciferase signal decay, presumably as OVA-expressing cells are cleared from the site by cytotoxic OVA (ovalbumin)-specific CD8+ T cells. A second booster vaccine administered 10 days after initial adoptive cell transfer (ACT) was effective in further recruiting transferred lymphocytes to the secondary vaccine site indicating persistence of functionally competent, genome-modified T cells (Fig. 2c). Although the s.d. vaccine appeared to be effective, we sought

alternative, more potent strategies to boost long-term engraftment and transgene expression.

**T-cell vaccine promotes engraftment of T cells**. To further augment engraftment and survival, we developed a vaccination approach relying on transposon-based genome modification of CD8+ T cells with the pTSB-CAG-OVA transposon construct instead of the s.d. vaccine. In this model we co-administered OVA-expressing OT-1 T lymphocytes intravenously (i.v.) at a 1:20 ratio to the therapeutic effector cells (Fig. 3a). We hypothesized that this unique vaccination strategy would produce a stable lymphotropic cellular vaccine that would provide prolonged OT-1 T-cell stimulation and augment engraftment and persistence[18]. Indeed, we observed a significantly enhanced engraftment of OT-1-specific cells transferred concomitantly with the i.v. OVA T-cell vaccine compared to cells transferred and stimulated with two consecutive s.d. plasmid vaccines, as determined by whole animal luminescence (Fig. 3b). We observed an increase in luciferase expression as analyzed by area under the curve when comparing the i.v. cellular vaccine to the s.d. vaccine (Fig. 3c, d, 65% increase, $p = 0.033$) over a period of 60 days. Therefore, we continued to use the cellular vaccination approach for in vivo long-term transgene expression from antigen-specific T cells.

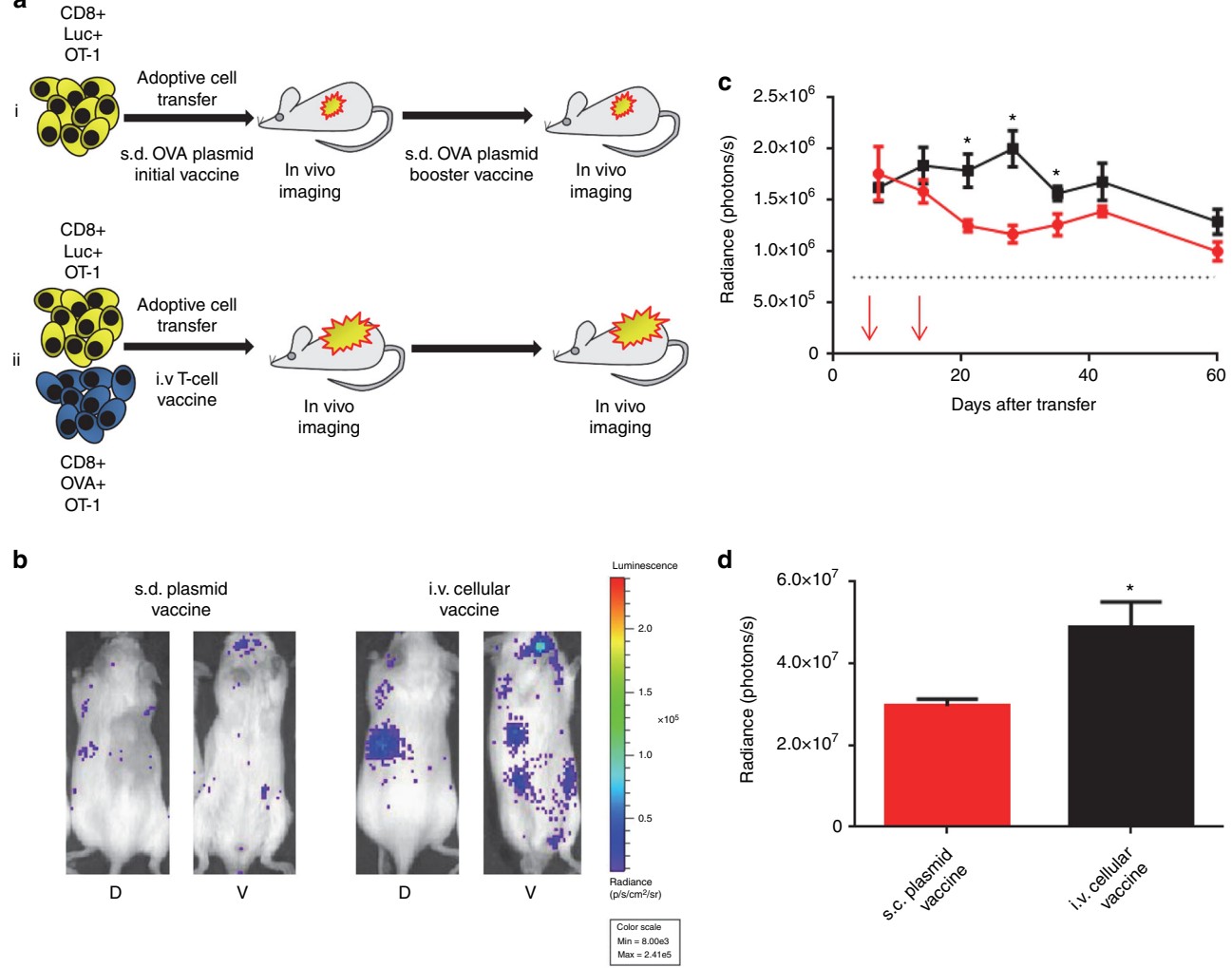

**Fig. 3** A comparison of vaccination approaches for augmenting T-cell engraftment. Vaccination approaches for augmenting lymphocyte engraftment were evaluated by infusing $1 \times 10^7$ CD8+ T cells genome-modified with the pT-effluc-thy1.1 transposon. **a** Schematic representation of experiments comparing (i) subdermal vaccination (s.d.) and (ii) intravenous (i.v.) vaccination; yellow cells indicate T cells transfected with pT-effluc-thy1.1 and pCMV-m7pB, blue cells indicate vaccine T cells transfected with pTSB-CAG-OVA and pCMV-SB100X. **b** Representative bioluminescent images taken 21 days after iv transfer of $1 \times 10^7$ genome-modified CD8+ T cells comparing the engraftment facilitated by a subdermal plasmid vaccine regimen (left panels) or an i.v. OT-1 T-cell vaccine (D, dorsal; V, ventral) ($N = 5$). **c** Plot showing total (dorsal + ventral) luciferase signals measured in mice that received either two subdermal plasmid vaccines at days 1 and 10 after transfer (red arrows), or a single i.v. OT-1 T-cell vaccine concomitantly during autologous transfer. Dashed line indicates mean luminescence signal observed in mice injected with saline without d-luciferin ($N = 5 \pm$ SEM, $*p < 0.05$ using Student's T-test comparing at given time points). **d** Plot showing area under the curve for the sum total of the measurements presented in **c** ($N = 5 \pm$ SEM, $*p < 0.05$ using Student's T-test)

**T-cell booster vaccine promotes persistence of T cells**. To produce a greater magnitude of expansion and persistence of the adoptively transferred T cells, we evaluated the utility of repeated boosts with a T cell-OVA vaccine. For these studies we utilized wild-type (WT) C57/Bl6 T lymphocytes transiently transfected ex vivo with an OVA antigen expression construct (pTSB-CAG-OVA) to provide a booster vaccination. The T cell-OVA vaccine was transferred by intraperitoneal (i.p.) injection to WT C57/Bl6 mice that had previously received i.v. infusions of pT-effluc-thy1.1-modified OT-1 cells in combination with OT-1-OVA + T cells as described above (Fig. 4a). Bioluminescent imaging indicated that this vaccination strategy produced robust recruitment of genome-modified adoptively transferred OT-1 lymphocytes to the vicinity of the injection site. Importantly, this vaccination approach facilitated potent re-boosting of adoptively transferred cells upon repeated administration (Fig. 4b). In mice

that underwent i.v. T-cell vaccine at the time of ACT, we observed efficient boosting of transferred cells by the i.p. cellular vaccine approach as long as 300 days after initial transfer (Fig. 4c). This indicates that transgene expression from antigen-specific T cells that were administered using our approach can be re-boosted at least 300 days after adoptive transfer.

**Long-term delivery of EPO by adoptively transferred T cells**. To determine if antigen-specific T cells could mediate therapeutic protein delivery in vivo, we sought to overexpress EPO in WT mice with a normal hematocrit. A raise in hematocrit would mean that sufficient EPO was expressed to increase hematocrit above normal levels. We engineered a murine EPO-expressing transposon (pT-EF1α-mEPO (Fig. 1)) and tested its ability to raise the hematocrit in mice following hydrodynamic tail vein

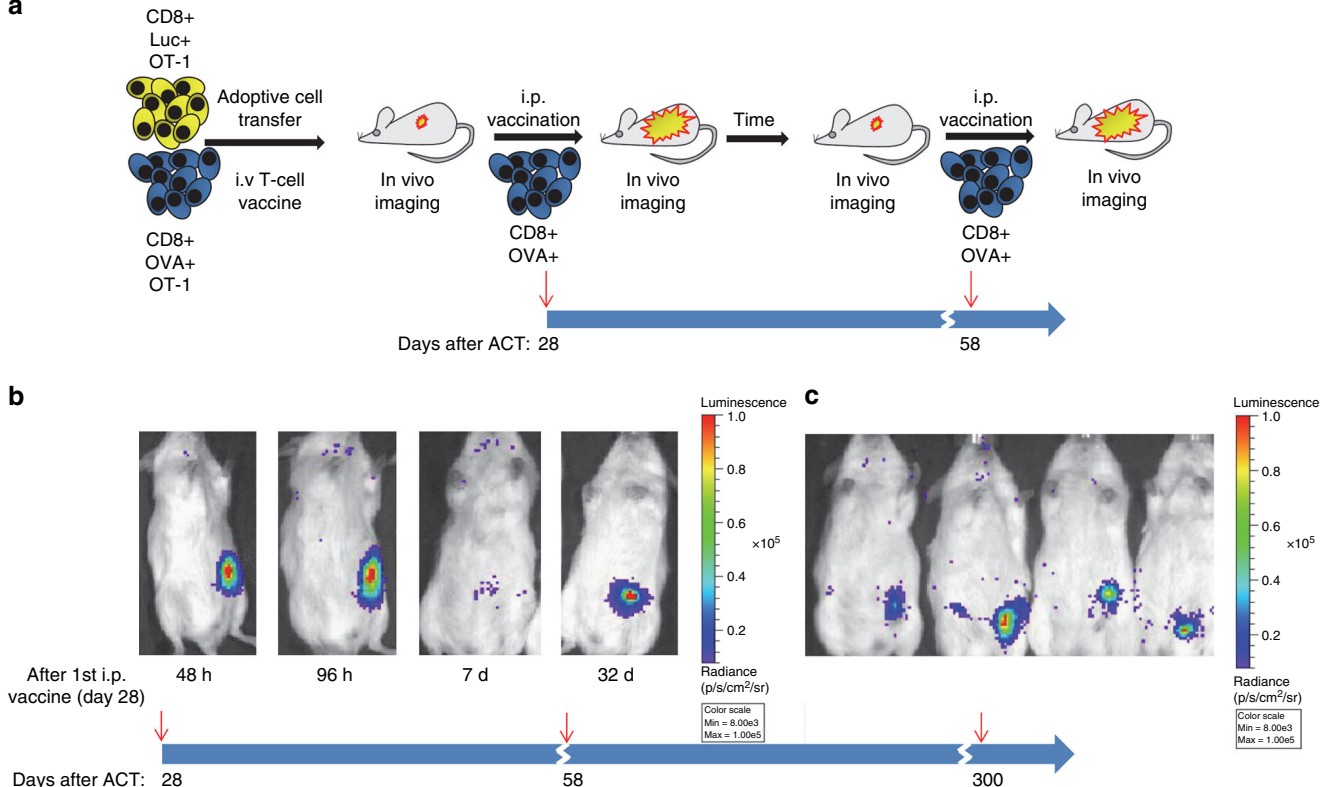

**Fig. 4** Cellular vaccine supports effective boosting and re-boosting. **a** Schematic representation of experimental protocol assessing cellular vaccine efficacy. **b** i.p. booster vaccines were administered to mice receiving transfer of $1 \times 10^7$ pT-effluc-thy1.1-modified CD8+ T cells. The blue timeline arrow indicates the days after ACT. Red arrows indicate i.p. cellular vaccine administration on days 28, 58, and 298. A representative image (**a–c**) corresponds to the blue arrow timeline ($N = 5$). **c** Effective boosting by i.p. T-cell vaccine was observed 300 days (image, **c**, shown are four of the five mice) after ACT of $1 \times 10^7$ pT-effluc-thy1.1-modified cells co-administered with the i.v. OVA + OT-1 T-cell vaccine

injection. As expected, severe polycythemia resulted as the hematocrit rose to over 80% (Supplementary Fig. 1). We next used a tetracycline-inducible expression system to express mEPO from mouse liver following hydrodynamic tail vein injection. Although this inducible system was able to prevent severe polycythemia, there was some expression leak indicated by an elevation in hematocrit observed prior to doxycycline administration (Supplementary Fig. 1). Although hydrodynamic tail vein injection allowed us to test our EPO transposons in vivo, clinical translatability of this approach to humans is unclear.

We next evaluated if our strategy of using antigen-specific T cells in vivo for therapeutic protein delivery by transferring EPO-expressing T cells to WT mice (Fig. 5a). For these studies we harvested OT-1 lymphocytes and genome-modified them to stably produce mEPO from the constitutive EF1α-promoter (Fig. 1) before adoptively transferring the cells into adult mice. We took advantage of our T-cell vaccine approach to establish the feasibility of producing EPO and raising the hematocrit in vivo. The amount of mEPO delivered by transposon-modified T lymphocytes was determined by monitoring plasma mEPO concentration following adoptive transfer (Fig. 5b). We transferred $2 \times 10^7$ OT-1 lymphocytes in by tail vein injection in conjunction with the i.v. OT-1/OVA T-cell vaccine. We observed a mean plasma mEPO concentration of 4405 ng/ml 24 h after adoptive transfer (compared to a mean plasma concentration of 193 ng/ml in untreated mice). Four weeks after ACT plasma mEPO concentration falls to a mean of 546 ng/ml. In order to determine the plasma mEPO concentration elicited by the i.p. cellular vaccine we administered the vaccine to on day 41 after ACT and observed a mean plasma

mEPO concentration of 2312 ng/ml 24 h after vaccination. Importantly, our infusion of mEPO-producing T cells resulted in an elevation of hematocrit (HCT) in WT (non-anemic) mice measured at 42 days (Fig. 5c).

We next evaluated if we could increase hematocrit beyond 6 weeks in WT mice. We transferred $2 \times 10^7$ OT-1 lymphocytes in by tail vein injection in conjunction with the i.v. OT-1/OVA T-cell vaccine. On days 7 and 14 after ACT we performed i.p. T-cell vaccination and monitored hematocrit periodically (Fig. 6a). We observed a 33% elevation in hematocrit by week 5 after transfer (Fig. 6b). The hematocrit remained significantly elevated above pre-transfer levels for more than 10 weeks without progressing to severe polycythemia. The i.p. cellular vaccination protocol was then repeated on weeks 12 and 13 effectively boosting hematocrit, which was maintained at an elevated state until at least 20 weeks after initial ACT (Fig. 6b). These results demonstrate the utility of using adoptively transferred antigen-specific T cells as a platform for sustained EPO delivery.

We extended our observations to determine if our T-lymphocyte-based method of EPO delivery could effectively reverse anemia of chronic kidney disease in the adenine nephrotoxicity model[19]. For this study mice were challenged with an adenine diet regimen, which lead to anemia (Fig. 6c). Anemic mice were then administered $2 \times 10^7$ EF1α-mEPO transposon-modified T lymphocytes and hematocrit was assessed every 2 weeks. Treated mice achieved hematocrit within the normal range (~42) by 2 weeks post ACT and maintained at these levels for at least 6 weeks (Fig. 6c). These results indicate that ACT with mEPO-modified T lymphocytes is an effective method of correcting anemia in an animal model.

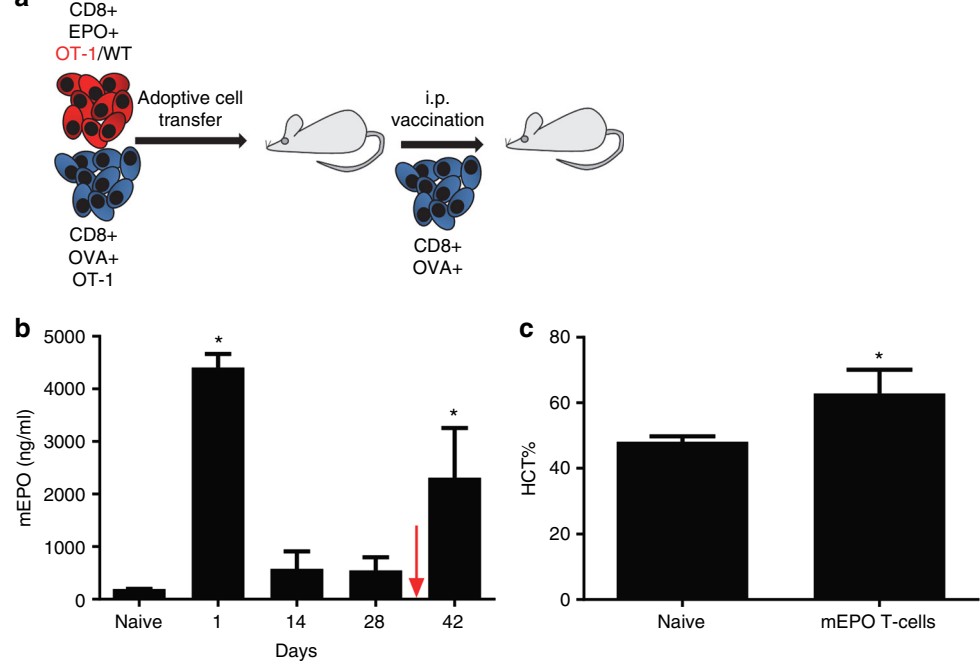

**Fig. 5** EPO delivery in vivo by EF1α-mEPO transposon-modified T lymphocytes. **a** Schematic representation of experiment evaluating ability of adoptively transferred transposon-modified T lymphocytes to produce EPO following adoptive cell transfer. **b** Graph of plasma EPO concentration in the days following ACT of $2 \times 10^7$ million EF1α-mEPO transposon-modified T lymphocytes. Statistical analysis was performed by comparing two pre-treatment levels using one-way ANOVA with multiple comparisons ($N = 5 \pm SEM$). Red arrow indicates administration of cellular vaccine at day 41. **c** Graph comparing hematocrit 6 weeks after ACT (mice from **b**) to that of untreated control mice. The mean hematocrit of treated mice was $62.8 \pm 3.2$ ($N = 5 \pm SEM$) and untreated mice was $47 \pm 1.4$ ($N = 6 \pm SEM$), $*p < 0.05$ using Student's $T$-test

**EBV-specific human T cells for regulated production of hEPO.**
We evaluated human T lymphocytes to determine if they could express and secrete therapeutic protein using EPO as a model system. The ability of engineered antigen-specific mouse T cells to express EPO in vivo and to raise hematocrit long term led us to examine the feasibility of modifying human antigen-specific T cells to express EPO. Having previously used the *piggyBac* transposon system to genome modify EBV-specific T cells[13], we asked whether EPO-modified EBV-specific T cells would retain antigen specificity. In consideration of possible future clinical application, we aimed to develop and inducible system for regulation of therapeutic protein. In the case of EPO expression, a system that allows for transgene expression to be shut off in the case of elevated hematocrit would be ideal. A variety of systems have been developed to allow for drug-inducible gene expression[20]. In these studies we evaluated the potential for regulated expression of recombinant human EPO in EBV-specific T cells using the tetracycline response system[21, 22]. EBV-specific lymphocytes, generated as previously described[23], were genome-modified with the pT-tight-hEPO transposon (Fig. 1). EPO production and secretion by the EBV-specific T cells in the presence and absence of tetracycline was monitored by quantifying culture media EPO content using an enzyme-linked immunosorbent assay (ELISA). We observed similar levels of EPO in the culture supernatants from mock-transfected EBV lymphocytes and from pT-tight-hEPO-modified lymphocytes in the absence of added tetracycline. However, we observed robust induction of EPO secretion into the cell culture supernatants from pT-tight-hEPO-modified cells in the presence of 1 µg/ml tetracycline (Fig. 7a). We evaluated hEPO expression by western blot comparing hEPO produced from a human T cell line to that of Chinese hamster ovary (CHO) cells, which have previously been used to produce EPO and other therapeutic proteins[24]. We found

that transfection of the same hEPO-producing vector (pT-tight-hEPO) into human Jurkat T cells when compared to CHO cells resulted in a different migration upon gel electrophoresis (Supplementary Fig. 2). Therefore, we infer that post-translational modification of hEPO produced from human T cells may differ from that in other cell types. How these modifications might affect clinical use remains to be determined. Nonetheless, our results indicate that the Tet-ON tetracycline response system allows for tightly regulated production and secretion of EPO in EBV-specific T cells.

Previous reports have demonstrated the long-term persistence of genome-modified EBV-specific T cells in vivo revealing them to be a possible long-term cellular delivery vehicle for peptide hormones such as EPO[25]. The utilization of EBV-specific lymphocytes for EPO delivery is predicated on the idea that these cells retain target specificity after in vitro expansion and genome modification. Using a lactate dehydrogenase (LDH) release cytotoxicity assay, we showed that EPO-modified EBV-specific T cells do indeed retain target specificity (Fig. 7b), killing autologous, but not human leukocyte antigen-mismatched EBV-transformed B lymphoblastoid cell lies (LCLs). Therefore, human antigen (EBV)-specific T cells can be genome-modified to express EPO and retain their antigen specificity.

**Discussion**
We sought to determine if antigen-specific T cells could be used as a cellular vehicle for long-term peptide hormone therapy. We used EPO as our peptide hormone of choice given the ease of measurement of biological readout of EPO levels raising hematocrit in vivo. We used the *piggyBac* transposon system for non-viral genetic modification of T cells because of its proven use in human T cells and other clinically relevant cell populations[26]. We

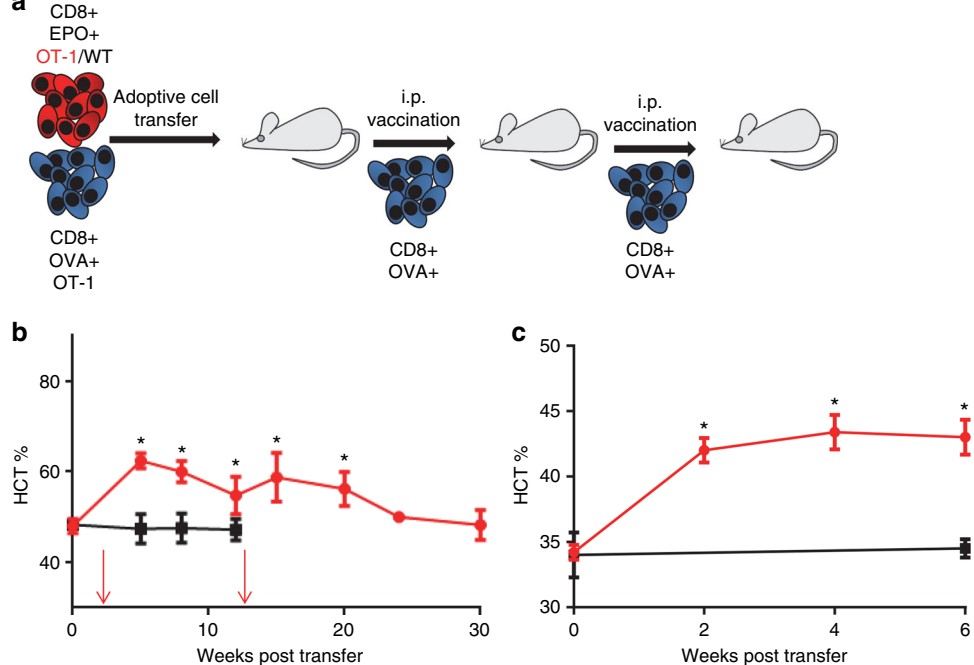

**Fig. 6** EF1α-mEPO transposon-modified T lymphocytes raise hematocrit in vivo. **a** Schematic representation of experiment evaluating ability of adoptively transferred transposon-modified T cells to raise hematocrit in WT C57Bl6 mice. Red cells indicate OT-1 (red) or wild-type CD8+ T cells transfected with pT-ef1α-mEPO and pCMV-m7pB. **b** Plot of mean hematocrit measurements observed in mice that received $2 \times 10^7$ pT-EF1α-mEPO genome-modified wild-type T cells (black) or OT-1 T cells (red), and an i.v. OT-1 T-cell vaccine. Additional i.p. T-cell vaccines were administered (red arrow) on days 7 and 14, and then again on days 84 and 91 (OT-1 group) after initial transfer. Statistical analysis of hematocrit measurements were performed by comparing to pre-ACT levels using the Student's T-test ($N = 5 \pm$ SEM, *$p \leq 0.01$). The mean hematocrit for untreated mice was $47 \pm 1.4$ ($N = 6 \pm$ SEM). **c** Plot of mean hematocrit measurements of mice subjected to adenine-induced anemia. Mice treated with EF1α-mEPO transposon-modified T lymphocytes achieved a mean hematocrit > 42 by 2 weeks post ACT compared to a mean hematocrit of 34 for the untreated group. $N = 5 \pm$ SEM for ACT-treated (red) and $N = 3 \pm$ SEM for untreated (black). *$p < 0.05$ using the Student's T-test comparing treated to untreated groups at week 0 and 6

demonstrated the ability to perpetuate the persistence of transposon-modified antigen-specific mouse T cells in vivo using vaccination, and then used this strategy to raise the hematocrit long term in vivo via EPO transgene expression. We then extended our study by demonstrating that human antigen (EBV)-specific T cells could be modified to inducibly express EPO and retain their antigen specificity.

A previous report demonstrated delivery of hEPO using antigen-specific B lymphocytes using Eμ- and Igλ-based hEPO-expressing transgenic mice[27]. Although a proof-of-principle approach using B cells, this involved generating transgenic mice expressing hEPO from the Eμ and Igλ loci. In contrast to our approach, this approach did not involve genetic modification of mature cells that were adoptively transferred. To our knowledge, this approach has not yet been successfully translated to human cells. To determine if T cells were a viable EPO delivery platform, we first verified that they could undergo genome modification and stably secrete functional hormone after adoptive transfer. We utilized the OT-1 transgenic TCR model for these studies because these cells harbor predetermined antigen specificity providing us with the opportunity to elicit TCR engagement facilitating amplification and stable engraftment in vivo by various vaccination models. During the course of these studies we identified an effective cellular vaccine approach, which takes advantage of several unique characteristics of T cells. Previous studies have suggested that T lymphocytes modified to express specific antigen can augment CD8+ T-cell responses in cancer xenograft models[18]. We sought to extend these studies using T cells as the cellular vaccine platform with the ultimate goal of peptide hormone delivery. Indeed, we observed augmented engraftment and

persistence of T cells when ACT preparations were "spiked" with a vaccine of stably genome-modified, OVA antigen-expressing OT-1 CD8+ T cells. This enhanced engraftment coupled with an i.p. cellular vaccination booster strategy provided sufficient EPO-modified T-cell engraftment and persistence to affect hematocrit for up to 4 months after ACT. Previous research supports the notion that cross presentation may be the primary mechanism by which T cells modified to express antigenic peptides provide effective vaccination[28].

The key to delivering therapeutic proteins long term via cell therapy is to facilitate efficient, stable, long-term engraftment of protein-producing cells. We chose to evaluate EBV-specific CD8 + T cells as a platform for EPO delivery because these cells can be efficiently expanded in vitro and because they respond to antigenic stimulation by latent viral infection in vivo undergoing expansion and resulting in stable persistence in vivo[3, 4]. Our results demonstrate the feasibility of modifying in vitro-expanded EBV lymphocytes to produce EPO under the tight control of a tetracycline response element without compromising the functional target specificity required to facilitate engraftment and persistence after autologous transfer into a host with latent EBV infection. A previous report has demonstrated the development of autoimmune anemia in a non-human primate model following EPO gene therapy using adeno-associated viral vector delivery[11]. Future studies will be necessary to determine if non-viral piggy-Bac-mediated modification of T cells to produce EPO carries the same risk.

Genetically modified antigen-specific T cells represent an attractive platform for long-term cellular delivery of therapeutic peptides. Antigen-specific T cells are more terminally

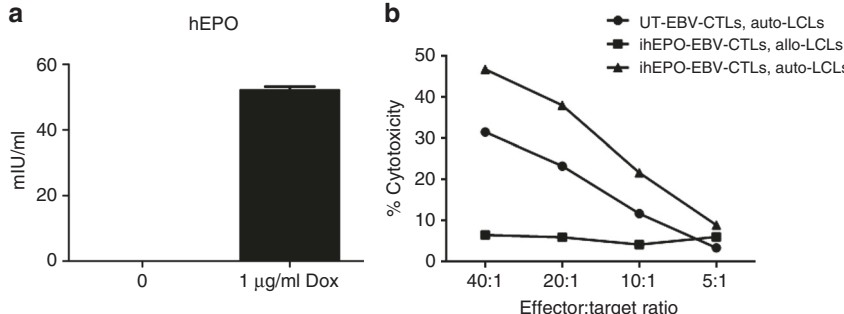

**Fig. 7** Modified EBV-specific T lymphocytes produce EPO and retain target specificity. **a** Genome modification of EBV-CTLs with pT-tight-hEPO results in tight regulation of EPO expression ($N = 3 \pm$ SEM). **b** pT-tight-hEPO-modified EBV-CTLs lyse target cells with similar efficiency as unmodified EBV-CTLs, indicating that genome-modified EBV-CTLs retain target specificity. Untransfected EBV-CTLs (UT-EBV-CTLs) lysed autologous LCLs (auto-LCLs) as did EBV-CTLs transfected with pT-tight-hEPO (ihEPO-EBV-CTLs) with auto-LCLs. However, ihEPO-EBV-CTLs did not lyse allologous LCLs (allo-LCLs)

differentiated than stem cell populations making them a safer alternative for genome modification. The T-cell population can be re-boosted via vaccination with cognate antigen resulting in re-boosting of protein production. Alternatively, an inducible expression system can be used to more tightly regulate protein production from these cells as we demonstrated herein with human T cells. The safety of such cells could by improved through co-administration of an inducible suicide gene[29], which can also be introduced into human T cells using the *piggyBac* system as we have previously demonstrated[13]. Insertional mutagenesis remains a risk when using *piggyBac* to modify cells for therapeutic application. Chimeric transposases targeting user-defined chromosomal elements can alleviate this risk, though this technology remains in its infancy for targeting integration with transposons[30]. Thus far, our results indicate that antigen-specific T cells represent a viable platform for long-term therapeutic delivery of EPO and possibly other peptide hormones.

## Methods
**Vectors**. The plasmid vectors pT-effluc-thy1.1[31], pCMV-m7pB[32], and pCMV-SB100X[32] were described previously. The pT-EF1α-mEPO was synthesized by Cyagen Biosciences (Santa Clara, CA). The pTSB-CAG-OVA vector to encode a kozak consensus sequence followed by the nine amino-acid peptide sequence corresponding to chicken ovalbumin $_{257-264}$ and a stop codon (5′-GATCGC-CACCATGAGTATAATCAACTTTGAAAAACTGTAACCGG-3′) by replacing the *piggyBac* terminal repeats (TRs) in pT-CAGLuc[33] with the *Sleeping Beauty* TRs and swapping out the luciferase cDNA for the OVA peptide. The vector pT-Tight-hEPO was generated by blunt cloning the Tet-ON element from pT-Tet-ON (Clontech, Mountain View, CA) into the multiple cloning site (MCS) of the zeo-pT-MCS vector[31] to make zeo-pT-MCS-Tet-ON. The TRE-hEPO element was cloned into the *Xho*I site of zeo-pT-MCS-Tet-ON generating TRE-hEPO-MCS-Tet-ON. We then synthesized the 250 bp core insulator sequence from the chicken B-globin 5′HS4 element with flanking *Asc*I and *Age*I sites (Genescript, NJ). This was then cloned into the MCS of the zeo-pT-TRE-hEPO-MCS-Tet-ON vector to generate pT-Tight-hEPO. All plasmid vectors were confirmed by DNA sequencing.

**Animals**. Eight- to ten-week-old WT C57/Bl6 mice, albino C57/Bl6 mice (B6-Tyr$^{c2j}$), and OT-1 TCR transgenic mice harboring T lymphocytes that recognize the MHC-I (H-2K$^b$)-restricted epitope of chicken ovalbumin$_{257-264}$ (SIINFEKL) were obtained from Jackson Labs. Anemia was induced by adenine diet as described[19]. Briefly, animals were fed LabDiet 5001 (St. Louis, MO) supplemented with 0.25% adenine ad libitum for 2 weeks followed by 1 week on normal adenine-free chow (LabDiet 5001, St. Louis, MO) and then one additional week on 0.25% adenine chow before maintaining mice on adenine-free chow for the duration of the study. All animal experiments were approved by the Institutional Animal Care and Use Committee of Vanderbilt University Medical Center.

**Adoptive transfers and vaccines**. Lymphocytes for adoptive transfer and cellular vaccine preparation were harvested from adult mouse spleens. Lymphocytes were isolated from whole spleen by crushing against 70 µm mesh with the back of a sterile 10 ml syringe plunger and washing once with phosphate-buffered saline (PBS) before pelleting cells by spinning at $300 \times g$ for 5 min. The cells were suspended in PBS and lymphocytes were isolated using lympholyte cell separation medium (Cedarlane, Burlington NC) according to the manufacturer's instructions.

CD8+ lymphocytes were purified from the mixed lymphocyte preparation using the MACS mouse CD8a+ T-cell isolation Kit (Miltenyi Biotec, Auburn, CA). Purified CD8+ lymphocytes were then expanded in complete T-cell medium (TCM; Advanced RPMI containing 10% fetal bovine serum) supplemented with interleukin (IL)-2 [10 ng/ml] (Peprotech, Rocky Hill, NJ) and anti-CD3e [2.5 µg/ml] (Fisher Scientific, Hampton, NH) for 3 days resulting in 7- to 10-fold expansion in cell numbers. After in vitro expansion the cells were washed briefly in PBS by centrifugation and transfected with the Neon (Life Technologies, Grand Island, NY) transfection system according to the manufacturer's instructions for mouse T cells as described previously[31]. Cellular vaccines were generated by transfecting in vitro-expanded T cells with a mixture of 5 µg pCMV-SB100X and 25 µg pTSB-CAG-OVA in the case of the i.v. vaccine and 25 µg pTSB-CAG-OVA in the case of the i.p. vaccine. The s.d. plasmid vaccines were prepared by mixing 5 µg pTSB-CAG-OVA and 1 µg pCMV-SB100X with in vivo-jetPEI (Polyplus, New York, NY) according to the manufacturer's instructions. Dorsal flanks were shaved and 30 µl of vaccine mixture was injected subdermally using an insulin syringe. In vitro-expanded OT-1 T cells were prepared for adoptive transfer by transfection with 5 µg pCMV-M7PB and 25 µg of pT-effluc-thy1.1 or 25 µg of pT-EF1a-mEPO.

Prior to adoptive transfer, mice were preconditioned by exposure to 5 Gy of lymphodepleting radiation using a cesium irradiator. T cells were transfected as described above, allowed to recover in complete TCM for 1.5 h and then adoptively transferred via tail vein injection. A volume of 30 µl of the plasmid vaccine mixture was transferred by s.d. injection to the shaved flank of recipient mice using a 28 gauge needle. Intravenous T-cell vaccines were prepared by transfecting CD8+ OT-1 T cells as described above mixed at a ratio of 19:1 with the pT-effluc-thy1.1 or pT-EF1α-mEPO, respectively, and co-administered during adoptive transfer. The i.p. T-cell vaccines were prepared by transfection of in vitro-expanded CD8+ T cells with 25 µg pTSB-CAG-OVA using the Neon transfection system, washed and suspended in PBS at $1 \times 10^6$ cells/ml, and 0.1 ml was immediately transferred into recipient mice by i.p. injection.

**Bioluminescent imaging**. Mice were anesthetized using isoflurane and injected i.p. with 100 µg luciferin substrate (Perkin Elmer, Waltham, MA) in PBS. Approximately 10 min after luciferase injections mice were imaged on the Xenogen IVIS 200 (Perkin Elmer, Waltham, MA) as described previously[22]. All data shown represent mean luminescence observed by summing dorsal and ventral measurements obtained from identical region of interests drawn over the trunk and head of each individual mouse.

**Assessment of hematocrit**. Blood was collected in Microvette CB300 LH Lithium-Heparin collection tubes (Sarstedt, Newton, NC) by saphenous vein bleeding. Hematocrit was measured using the FORCYTE Hematology Analyzer (Oxford Science, Oxford, CT).

**Expression of mEPO from mouse liver**. Plasmid DNA was delivered to mouse liver using constitutive or inducible mEPO-expressing *piggyBac* vectors (Supplementary Fig. 1) using hydrodynamic tail vein injection as described[22, 33]. Hematocrit was measured as described above.

**Quantification of EPO expression using ELISA**. Plasma mouse EPO levels were quantified by ELISA (R&D Systems, Minneapolis, MN) according to the manufacturer's instructions. Blood was obtained by saphenous vein bleed into heparinized Microvette CB300 LH Lithium-Heparin collection tubes (Sarsted) and plasma was harvested by immediately centrifuging blood at $1500 \times g$ for 5 min and stored at $-80$ °C. Human EPO concentration was determined using Human EPO Platinum ELISA kit (eBiosciences, Waltham, MA) according to the manufacturer's instructions. Plates were read on a Fluostar Omega plate reader (BMG Labtech, Cary, NC).

**Immunoaffinity isolation and western blot of human EPO**. CHO (American Type Culture Collection (ATCC) CCL-61) cells or human Jurkat T (ATCC TIB-152) cells were transfected with pT-tight-hEPO and treated with 1 mg/ml doxycycline for 24 h. Immunoaffinity isolation of EPO was used prior to western blot analysis as described by others[34]. One hundred microliters of each medium or 50 μl of standard (84 pg/ml of recombinant hEPO from Abcam (Cambridge, MA) were added to each well of an hEPO immunoaffinity isolation plate (StemCell Technologies, Vancouver, BC) and incubated at room temperature with shaking for 3 h then overnight without shaking at 4 °C. Wells were washed five times with PBS and proteins in each well solubilized in 25 μl 1× SDS-polyacrylamide gel electrophoresis loading buffer and run on a 4–12% NuPage Bis-Tris gels followed by transfer to nitrocellulose for immunoblotting. hEPO was detected using α-hEPO (R&D Systems, clone AE7A5, 1 mg/ml) at 1:1000 and 800 IRD-labelled donkey α-mouse (LI-COR, Lincoln, NE) at 1:15 000 using an Odyssey Infrared Imaging System (LI-COR). The uncropped blot is shown in Supplementary Fig. 2.

**LDH-release cytotoxicity assay**. Cytotoxicity measurements were performed using Cytotox-96 Kit (Promega, Madison, WI). Briefly, 10 000 EBV LCLs (target cells) were incubated with the indicated number of effector T cells for 4 h in a humidified incubator at 37 C in 5% $CO_2$. Lysis of the target cells by effectors leads to LDH release, which is measured by conversion of tetrazolium salt to a red formazan product detected at 490 nm on a Fluostar Omega plate reader.

**Flow cytometry**. Twenty-four hours after transfection 1 million cells were collected and stained with BB515-labeled anti-mouse thy1.1 antibodies (0.06 μg/test, BD564607, Fisher Scientific) and acquired on a LSRFortessa (BD Biosciences, San Jose, CA) cytometer.

**Genome modification of human EBV-specific T cells**. EBV LCLs were generated as previously described[35]. Peripheral blood mononuclear cells (PBMCs) were prepared from blood samples obtained from healthy donors under informed consent and approved by the Institutional Review Board of Baylor College of Medicine and Vanderbilt University Medical Center using Ficoll density centrifugation[36]. T cells were cultured in TCM containing Advanced RPMI (Invitrogen, USA) supplemented with 10% fetal calf serum and 2 mM L-GlutaMAX-1 (Invitrogen). PBMCs were incubated overnight with 10 ng/ml of IL-7 (eBiosciences, Fisher, Waltham, MA). Cells were harvested the next day and nucleofected with 5 μg of each vector using the Human T-cell Nucleofector Kit (Lonza, Basel, Switzerland) and Amaxa nucleofector device (program U-014, unstimulated T cells). The nucleofected PBMCs were rested overnight in complete TCM with 10 ng/ml IL-7 and 1000 U/ml of IL-4 (eBiosciences, Fisher). PBMCs were then stimulated with 40 Gy irradiated auto-LCLs at a ratio of 40:1 for 8 days. On day 11, cells were resuspended at a ratio of 4:1 with irradiated auto-LCLs. On day 20, ihEPO-CTLs were enriched by magnetic selection using a MACS human CD19+ cell enrichment kit according to the manufacturer's instruction on an LS column (Miltenyi Biotec).

**Sample Size**. For animal studies, five independently treated animals were evaluated unless the values were previously known (such as baseline EPO levels in naive mice). For in vitro experiments, at least three independent experiments were performed. Those performing experimental analysis were blinding during data acquisition.

**Data availability**. All relevant data are available from the authors upon reasonable request.

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

## Acknowledgements

This work was supported in part by National Institutes of Health grant DK093660 to M. H.W., Department of Veterans Affairs grant BX002190 to M.H.W., and the Vanderbilt O'Brien Kidney Center (NIH DK114809) and the Vanderbilt Institute for Clinical and Translational Research (NIH UL1TR002243). S.S. was supported in part by the HHMI Med into Grad Training Grant through the Baylor College of Medicine Translational Biology and Molecular Medicine Program. R.T.O. was supported by DK007569. Some core services were performed through Vanderbilt University Medical Center's Digestive Disease Research Center supported by NIH grant P30DK058404.

## Author contributions

R.T.O., S.S., R.A.V., R.W., and L.E.W. performed experiments. M.H.W. and C.M.R. oversaw experiments. R.T.O. and M.H.W. wrote the paper with editing by S.S. and C.M.R.

## Additional information

**Competing interests:** The authors declare no competing interests.

