## [Peer Review File · Nature Communications]

Reviewers' comments:

Reviewer #1 (Remarks to the Author):

The paper by O'Neil and colleagues reports a novel approach to deliver a protein therapeutic by infusion of genetically modified mouse and human lymphocytes. This is an exciting approach that has many practical and translational applications. Authors should address the below:

1. In the syngeneic mouse system, OT-1 T cells expressing mEPO raised the HCT in mice. Authors should show levels of murine EPO in the serum of these mice and indicate the range of variation and durability of expression of EPO in the serum.

2. Authors should discuss that transgenically encoded epo can lead to loss of tolerance and autoimmunity in nonhuman primates: Gao G et al, Erythropoietin gene therapy leads to autoimmune anemia in macaques. *Blood*. 2004;103(9):3300-2.

3. Authors should reference the patent literature that describes vaccination to boost genetically modified lymphocytes encoding protein hormones:

Publication number WO2000057920 A2

Publication type Application

Application number PCT/GB2000/001225

Publication date Oct 5, 2000

4. Line 205 page 9: remove sentence: "this sentence seems a bit redundant? Maybe not necessary...maybe combine with following sentence?".

Prof. Carl June

Reviewer #2 (Remarks to the Author):

The authors utilize a system using T-cells to produce a therapeutic protein (erythropoietin) stimulated for expansion and extended survival by vaccination with their cognate antigen (OT-1) or EBV. This is a novel approach to deliver a therapeutic protein, and has wide potential application. The T-cells can be stimulated to expand and persist by exposure to their cognate antigen, thus allowing prolonged production of the therapeutic protein. Genetically modified antigen-specific T cells represent an attractive platform for long-term cellular delivery of therapeutic peptides.

Critique:

The system requires a controllable production of the therapeutic gene product. In this case, polycythemia occurred. An inducible tetracycline induced system was tested, but a mechanism to down regulate the therapeutic protein expression is also required.

The primary goal of the study was to use this approach to induce secretion of a therapeutic protein, in this case, erythropoietin. The production of EPO should not be a supplemental figure, but should be included in the primary manuscript.

The ability to regulate expression of the therapeutic protein is also central to this therapeutic strategy and the use of the tetracycline inducible expression system should also be included in the primary manuscript and not a supplemental figure.

Minor:

The manuscript includes internal comments among the authors – line 2

Summary:

Use of T-cells with vaccination or stimulation through an endogenous cognate antigen, is a novel effective mechanism of prolonged therapeutic protein production. This is an important proof of principle which merits publication.

Reviewer #3 (Remarks to the Author):

O'Neil et al present a novel concept using the PiggyBac transposon (rather than viral vectors) for modifying T lymphocytes to secrete therapeutic proteins.

The manuscript is well written with a well-designed pre-clinical work.

The authors should perform a thorough proof reading as it seems some parts are not completely finalized (e.g. page 9).

Though clinical application of the described technology will require ample pre-clinical work to compile a pre-clinical package which will support Phase I studies in human, some questions arise as to the feasibility of translating the pre-clinical findings into clinical application. I believe that the authors should consider such questions and discuss them in the manuscript and taking them into account towards further research.

A major issue in using genetically modified cells for secretion of therapeutic proteins is dosing. In vitro dosing prior to implantation of transduced cells may be challenging based on the unknown engraftment potency post administration and actual in-vivo secretion of the protein. In the case of EPO (as will be with other types of protein) there is a known toxicity (specifically cardiovascular in the case of EPO) in supra-physiologic levels. Authors have not shown the levels of EPO in mice serum post administration of transduced cells, this is extremely important since we know that high levels of EPO may be needed in order to increase Hct especially in wild type animals. On the other hand, physiological mechanisms exist to regulate serum EPO (e.g. soluble EPO receptor), and it may well be that albeit high levels secreted, the overall serum EPO will be within the desired range. This issue must be evaluated.

One of the caveats of gene therapy at large is insertional mutagenesis, though thought to be less prevalent with the PiggyBac system, this has to be evaluated as a major part of the method's safety in human.

Did authors evaluate the sequence of EPO secreted from the modified T cells? It is important to evaluate if the sequence is same as autologous EPO especially regarding post translation modifications. This may have an effect on EPO effectiveness.

It is also important to evaluate whether anti EPO antibodies have been generated against EPO secreted from modified cells. This is of high importance if treatment is to be given to individuals who only need a boost of EPO but still secrete autologous EPO, this is not only a matter of treatment efficacy. This may be true for other types of protein to be secreted via modified T cells. Where there any immunogenicity signs in the treated animals?

Response to reviewers:

We thank the reviewers for their helpful comments. We have revised the manuscript and have included new data. We hope that the paper is now acceptable for publication. All changes in the manuscript are highlighted as requested.

Reviewer #1:

The paper by O'Neil and colleagues reports a novel approach to deliver a protein therapeutic by infusion of genetically modified mouse and human lymphocytes. This is an exciting approach that has many practical and translational applications. Authors should address the below:

1. In the syngeneic mouse system, OT-1 T cells expressing mEPO raised the HCT in mice. Authors should show levels of murine EPO in the serum of these mice and indicate the range of variation and durability of expression of EPO in the serum.

We have now demonstrated murine EPO levels in new figure 5.

2. Authors should discuss that transgenically encoded epo can lead to loss of tolerance and autoimmunity in nonhuman primates: Gao G et al, Erythropoietin gene therapy leads to autoimmune anemia in macaques. Blood. 2004;103(9):3300-2.

We have included this in the discussion. Line 317..." A previous report has demonstrated the development of autoimmune anemia in a non-human primate model following EPO gene therapy using adeno-associated viral vector delivery (11). Future studies will be necessary to determine if non-viral modification of T cells to produce EPO carries the same risk."

3. Authors should reference the patent literature that describes vaccination to boost genetically modified lymphocytes encoding protein hormones:

Publication number WO2000057920 A2

Publication type Application

Application number PCT/GB2000/001225

Publication date Oct 5, 2000

We have added this to the introduction. Line 63... "Vaccination can be used to boost genetically modified lymphocytes encoding protein hormones (5)."

4. Line 205 page 9: remove sentence: "this sentence seems a bit redundant? Maybe not necessary...maybe combine with following sentence?".

We have removed this sentence that was unintentionally left from our editing process.

Prof. Carl June

Reviewer #2:

The authors utilize a system using T-cells to produce a therapeutic protein (erythropoietin) stimulated for expansion and extended survival by vaccination with their cognate antigen (OT-1) or EBV. This is a novel approach to deliver a therapeutic protein, and has wide potential application. The T-cells can be stimulated to expand and persist by exposure to their cognate antigen, thus allowing prolonged production of the therapeutic protein. Genetically modified antigen-specific T cells represent an attractive platform for long-term cellular delivery of therapeutic peptides.

Critique:

The system requires a controllable production of the therapeutic gene product. In this case, polycythemia occurred. An inducible tetracycline induced system was tested, but a mechanism to down regulate the therapeutic protein expression is also required.

We first measured the ability of EPO produced from mouse T cells to raise hematocrit in wild type mice. This necessitates that polycythemia will occur. We have now added new data (Figure 6C) demonstrating correction of a mouse model of anemia out to 6 weeks raising an anemic hematocrit to the normal range using EPO producing mouse T cells *in vivo*.

The primary goal of the study was to use this approach to induce secretion of a therapeutic protein, in this case, erythropoietin. The production of EPO should not be a supplemental figure, but should be included in the primary manuscript.

We have now demonstrated murine EPO levels in new figure 5.

The ability to regulate expression of the therapeutic protein is also central to this therapeutic strategy and the use of the tetracycline inducible expression system should also be included in the primary manuscript and not a supplemental figure.

The supplemental figure shows the use of a tetracycline inducible system expressing EPO from mouse liver after gene transfer to liver (not T cells). We show this as hydrodynamic tail vein injection is commonly used for gene transfer in mice. Given that this figure does not involve T cells and supplements the T cell data, we prefer to leave it as a supplemental figure.

Minor:

The manuscript includes internal comments among the authors – line 2

We have removed these comments.

Summary:

Use of T-cells with vaccination or stimulation through an endogenous cognate antigen, is a novel effective mechanism of prolonged therapeutic protein production. This is an important proof of principle which merits publication.

Reviewer #3:

O'Neil et al present a novel concept using the PiggyBac transposon (rather than viral vectors) for modifying T lymphocytes to secrete therapeutic proteins.

The manuscript is well written with a well-designed pre-clinical work.

The authors should perform a thorough proof reading as it seems some parts are not completely finalized (e.g. page 9).

We have corrected this.

Though clinical application of the described technology will require ample pre-clinical work to compile a pre-clinical package which will support Phase I studies in human, some questions arise as to the feasibility of translating the pre-clinical findings into clinical application. I believe that the authors should consider such questions and discuss them in the manuscript and taking them into account towards further research.

A major issue in using genetically modified cells for secretion of therapeutic proteins is dosing. In vitro dosing prior to implantation of transduced cells may be challenging based on the unknown engraftment potency post administration and actual in-vivo secretion of the protein. In the case of EPO (as will be with other types of protein) there is a known toxicity (specifically cardiovascular in the case of EPO) in supra-physiologic levels. Authors have not shown the levels of EPO in mice serum post administration of transduced cells, this is extremely important since we know that high levels of EPO may be needed in order to increase Hct especially in wild type animals. On the other hand, physiological mechanisms exist to regulate serum EPO (e.g. soluble EPO receptor), and it may well be that albeit high levels secreted, the overall serum EPO will be within the desired range. This issue must be evaluated.

We have now demonstrated murine EPO levels in new figure 5.

One of the caveats of gene therapy at large is insertional mutagenesis, though thought to be less prevalent with the PiggyBac system, this has to be evaluated as a major part of the method's safety in human.

We agree and have added the following text starting at line 330... "Insertional mutagenesis remains a risk when using piggyBac to modify cells for therapeutic application. Chimeric transposases targeting user-defined chromosomal elements can alleviate this risk, though this technology remains in its infancy for targeting integration with transposons (35)."

Did authors evaluate the sequence of EPO secreted from the modified T cells? It is important to evaluate if the sequence is same as autologous EPO especially regarding post translation modifications. This may have an effect on EPO effectiveness.

We have added new data and the following text starting at line 249... "We evaluated hEPO expression by Western blot comparing hEPO produced from a human T cell line to that of Chinese hamster ovary (CHO) cells which have previously been used to produce EPO and other therapeutic proteins (24). We found that transfection of the same hEPO producing vector (pT-tight-hEPO) into human Jurkat T cells when compared to CHO cells resulted in a different migration upon gel electrophoresis (Supplemental Figure 2). Therefore, we infer that post-translational modification of hEPO produced from human T

cells may differ from that in other cell types. How these modifications might affect clinical use remains to be determined.”

It is also important to evaluate whether anti EPO antibodies have been generated against EPO secreted from modified cells. This is of high importance if treatment is to be given to individuals who only need a boost of EPO but still secrete autologous EPO, this is not only a matter of treatment efficacy. This may be true for other types of protein to be secreted via modified T cells. Where there any immunogenicity signs in the treated animals?

We did not observe any signs of immunogenicity in animals as we were able to increase hematocrit in wild type mice out to 20 weeks (Figure 6B). We have also added new data demonstrating the normalization of hematocrit in an anemic mouse model (adenine nephrotoxicity) out to 6 weeks (Figure 6C).

Reviewers' Comments:

Reviewer #2:

Remarks to the Author:

The authors have successfully responded to the review. I recommend acceptance

Reviewer #3:

Remarks to the Author:

My comments and questions were fully addressed and answered. The manuscript is to my opinion now eligible for publication.